# Malnutrition, enamel defects, and early childhood caries in preschool children in a sub-urban Nigeria population

**Morenike Oluwatoyin Folayan**[1]*, **Maha El Tantawi**[2], **Ayodeji Babatunde Oginni**[3], **Michael Alade**[4], **Abiola Adeniyi**[5], **Tracy L. Finlayson**[6]

**1** Obafemi Awolowo University, Ile-Ife, Osun State, Nigeria, **2** Faculty of Dentistry, Alexandria University, Alexandria, Egypt, **3** Innovative Aid, Abuja, Nigeria, **4** Obafemi Awolowo University Teaching Hospitals' Complex, Ile-Ife, Osun State, Nigeria, **5** Lagos State University College of Medicine, Lagos, Nigeria, **6** School of Public Health, San Diego State University, San Diego, California, United States America

* toyinukpong@yahoo.co.uk

**Data Availability Statement:** All relevant data are within the paper.

**Funding:** The research was not funded.

## Abstract

### Objectives

The study tried to determine if malnutrition (underweight, stunting, wasting, overweight) and enamel defects (enamel hypoplasia, hypomineralized second molar, amelogenesis imperfecta, fluorosis) were associated with early childhood caries (ECC). The study also examined whether malnutrition was associated with the presence of enamel defects in 0-5-year-old children.

### Methods

The study was a secondary analysis of primary data of a cross-sectional study assessing the association between maternal psychosocial health and ECC in sub-urban Nigerian population collected in December 2018 and January 2019. One hundred and fifty nine children were recruited. Exploratory variables were malnutrition and enamel defects. The outcome variables were the prevalence of ECC in 0-2-year-old, 3-5-year-old, and 0-5-year-old children. Multivariable Poisson regression analysis was used to determine the associations, and socioeconomic status, oral hygiene status, and frequency of in-between-meals sugar consumption were adjusted for. The adjusted prevalence ratios, 95% confidence intervals, and p values were calculated.

### Results

The prevalence of ECC was 2.1% in 0-2-year-old children and 4.9% in 3-5-year-old children. In adjusted models, underweight, stunting, and wasting/overweight were not significant risk indicators for ECC in either age group. 0-2-year-old children who had amelogenesis imperfecta (p<0.001) and fluorosis (p<0.001) were more likely to have ECC than were children who did not have these lesions. 3-5-year-old children who had hypoplasia (p = 0.004), amelogenesis imperfecta (p<0.001) and fluorosis (p<0.001) were more likely to have ECC than were children who did not have these lesions. 0-5-year-old children with hypoplasia

**Competing interests:** The authors have declared that no competing interests exist.

(p<0.001) and fluorosis (p<0.001) were more likely to have ECC than were children who did not have these lesions. There were significant associations between various types of malnutrition and various types of enamel defects.

## Conclusion

Although different types of malnutrition were associated with enamel defects, and enamel defects were associated with ECC, malnutrition was not associated with ECC. Further studies are needed to clarify the association between malnutrition and genetically and toxin-induced enamel defects.

## Introduction

Malnutrition—measured by three anthropometric variables for stunting, wasting, and underweight—is a major health problem for infants, toddlers and preschool children globally [1,2] and in Nigeria [3]. Nigeria is one of the top 10 countries with malnutrition of children under five years of age. The prevalence of childhood stunting, wasting, underweight and overweight in Nigeria is 37%, 7%, 22% and 2%, respectively [4]. Child growth failure–an outcome of malnutrition—increases the risk for physical, metabolic and cognitive problems in childhood, with impact on the cardiovascular and intellectual health in adulthood [5–8]. Growth failure is also associated with oral-health disorders, such as enamel defects [9].

Enamel defects are a group of disorders that includes enamel hypoplasia, hypomineralized second molar/molar incisor hypomineralization, amelogenesis imperfecta and fluorosis. Enamel hypoplasia is associated with malnutrition [10] due to the disturbance of ameloblastic activity during the secretory phase of amelogenesis [11]. The severity of the insult determines the extent of defect and the translucency of partially formed enamel [10, 11]. Moderate and acute malnutrition are likely to disturb ameloblastic activity [12]. Enamel hypoplasia is not only associated with chronic and acute malnutrition [13, 14], but also with bacterial and viral diseases [15, 16] and with very low birth weight [17], which are prevalent in Nigeria [18]. The prevalence of enamel hypoplasia in Nigeria was 4% in the primary dentition of pre-school children [19].

Based on population-based survey, the prevalence of hypomineralization of the second primary molar in Nigeria was 4.6% [20] and 5.8% in a school-based survey [21]. Precipitating environmental factors for hypomineralization include early childhood illnesses and prenatal maternal ill health [21], which are factors also associated with malnutrition. Borderline association between hypomineralized second primary molar and early childhood caries (ECC) in Nigeria also has been identified [22].

Studies on fluorosis have been largely limited to the permanent dentition, with reported global prevalence ranging between 6.7% and 32.2% [23] and few studies indicating association between caries and fluorosis in the permanent dentition [24, 25]. Caries is also associated with amelogenesis imperfecta in the permanent dentition [26]. There is a scarcity of studies about the association between fluorosis, amelogenesis imperfecta and ECC in Nigeria.

Factors associated with higher prevalence of ECC in Nigeria are poor oral hygiene and consumption of refined carbohydrate between meals more than three times a day [27, 28]. However, high sugar consumption was not associated with ECC when malnutrition was included in the analysis [28]. No study conducted in Nigeria has determined the association between malnutrition and ECC or the association between malnutrition and hypoplasia. This study

aimed to determine if malnutrition and enamel defects (enamel hypoplasia, hypomineralized second molar, amelogenesis imperfecta and fluorosis) are associated with ECC when oral hygiene and frequency of consumption of sugar are controlled for. The study also determined the association between malnutrition and enamel defects.

## Methods

This is a secondary data analysis of a population-based cross-sectional study, with data collected from 6-71-month-old children resident in the Ile-Ife, Central Local Government Area, Osun State, Nigeria, a sub-urban area. The Local Government has a population of 138,818, of whom 14,000 are children [29]. The primary data were collected between December 2018 and January 2019, to determine the association between ECC and maternal psychosocial factors [30].

### Study population

The study population for the primary study consisted of children aged 6–71 months living with their biological parents or legal guardians who gave written informed consent to participate in the study. Only child-mother dyads who were at home at the time of data collection were recruited for the study. The primary study data were reviewed, and children were excluded if they had medical conditions that could increase the risk of malnutrition and ECC, such as HIV infection [31].

### Sample size

Sample size for the primary study was calculated with the formula of Araoye [32], based on ECC prevalence = 6.6% [27], margin of error = 5%, and 95% confidence level. To recruit 95 children with ECC, the total sample size required was 1439.

### Sampling procedure

Study participants for the primary study were recruited through a household survey, which facilitated access to pre-school children, ensured appropriate distribution of socioeconomic status, and better represented children in the study area [33, 34]. The local government area was selected for the study to enable comparison with historical data that had been collected from the same population, using the same procedure, five years earlier [27].

A three-level, multi-stage cluster-sampling technique was used to identify eligible participants. In stage 1, 70 (10%) of the 700 enumeration areas were selected by balloting within Ife Central Local Government Areas designated by the National Populations Commission during the 2006 National census; 10% of the population is a representative sample for a household survey to give a 'snapshot' of the whole population [35]. For stage 2, every other household on each street in the eligible enumeration area was selected. Finally, in stage 3, respondents for interview and clinical examination were recruited. Only one eligible child and mother dyad in each household was included. The sample size for each enumeration site was determined by proportioning the study sample size per the population of the enumeration site. Recruitment of study participants continued until the required number of participants was reached.

### Data collection

The data for the primary study were collected electronically by trained and experienced field workers using interviewer-administered structured questionnaire on Open Data Kit–an online/offline platform for the collection and management of data. The questionnaire was

administered to the mothers. Five dentists conducted physical and clinical examinations. Kappa scores for intra-examiner reliability of dentists ranged between 0.80 and 0.92. A consultant pedodontist conducted the training and reliability testing and served as reference for the five dentists, who had inter-examiner reliability kappa scores of at least 0.82.

## Socioeconomic status

Socioeconomic status was assessed for its possible confounding effect of the relationship between malnutrition [36] and ECC. The index combines mother's education and father's occupation to obtain five socio-economic classes for children [37]. The classes were categorized into high, middle and low status [38].

## Cariogenic diet

Data on the frequency of consumption of sugary snacks between main meals was collected with the tool developed by Khami et al [27]. Consumption of sugary snacks between main meals three times a day or more is a significant risk factor for ECC [27].

## Nutritional status

Nutritional status was determined with the World Health Organization (WHO) AnthroPlus Software, which contains the WHO 2007 Reference for 5–19 years and the WHO Child Growth Standard for 0–5 years. The anthropometric measurements used in the study were height and weight [39], collected in line with the International Society for the Advancement of Kinanthropometry standard protocol [40]. Children whose height for age z-score was below minus two standard deviations (SD) from the median of the WHO reference population were considered stunted. Children whose weight for height z-score was below minus two SD from the reference population median were considered wasted. Children whose body mass index for age z-scores was below minus two SD from the reference population median were classified as underweight, while those with z-scores >2.00 were classified as overweight.

## Early childhood caries

ECC was defined as the presence of cavitated and non-cavitated lesions, filled or missing surfaces in any primary tooth in children less than 72 months of age [41]. The presence of ECC was determined with the dmft index based on the World Health Organization criteria [42]. The dmft score was obtained by adding the d, m and f scores for each child less than six years of age. The dmft score was dichotomized into 0 = ECC absent and >0 = ECC present.

## Oral hygiene status

Oral hygiene status was assessed with the index of Greene and Vermillion [43]. The index teeth and surfaces examined were the facial and lingual surfaces of teeth number 51, 55, 65, 71, 75, and 85. The debris and calculus scores were recorded, added, and divided by the number of surfaces examined to get the OHI-S score. Oral hygiene was considered "good" when the scores ranged from 0.0 to 1.2; "fair" when the scores ranged from 1.3 to 3.0; or "poor" when the score was 3.1 and above. For children who did not have the index teeth, all the teeth present were scored, and their average was calculated before being classified.

## Enamel defects

Hypomineralized primary second molar was identified when demarcated white, yellow or brown opacities more than or equal to 2 mm in diameter, were present on any of the surfaces

of the primary second molar [44, 45]. Fluorosis was identified when there was tooth mottling [46]. A diagnosis of enamel hypoplasia was made when there was either generalized deficiency of enamel formation, or localized deficiency seen as pits or grooves [47]. Amelogenesis imperfecta was identified when enamel hypoplasia and/or hypomaturation or hypocalcification randomly affected multiple teeth in no depictable chronological order [48].

### Data analysis

Descriptive statistics were provided for ECC presence (yes/no), malnutrition status (stunting, wasting, underweight and overweight), and enamel defects (hypoplasia, hypomineralized second primary molar, fluorosis and amelogenesis imperfecta). Multivariable Poisson regression models were used to assess the relationship between exposures (malnutrition status and enamel defects), confounders (socio-economic status, frequency of sugar consumption in-between-meals and oral hygiene) and the outcome variable (presence of ECC measured by prevalence ratio). We used robust variance estimation due to the sparse data on some variables. Explanatory variables were grouped into three blocks, and one block was introduced into the model at a time. Model 1 included the block of malnutrition status (stunted, underweight, wasted, overweight); Model 2 included the block of malnutrition status and enamel defects; Model 3 included variables from Model 2 and oral health practices associated with ECC, namely, frequency of daily consumption of sugar between meals and oral hygiene status, in addition to socio-economic status. Age was excluded because of possible collinearity, since it was used to compute malnutrition status. The models were adjusted for the cofounders, and the adjusted prevalence ratios (APR) were calculated. We tested for multicollinearity in each model, using variance inflation factor. A separate model was constructed to assess the association between the dependent variable (types of enamel defect) and independent variables (types of malnutrition)

Models were built for 0-2-year-old children, 3-5-year-old children in view of evolving evidence that ECC profile and risk indicators differ for these age groups [49]. A model was also built for and 0-5-year-old children since ECC is often assessed for the age group combined. Statistical analyses were conducted using Stata/SE 14.0 for Windows. The significance level was set at p≤0.05.

### Ethics approval

Ethical approval for the study was obtained from the Obafemi Awolowo University Teaching Hospitals Complex Health Research Ethics Committee (NHREC/27/01/2009a and IRB/EC/0004553).

### Results

A total of 1549 participants were recruited for the study. None of the participants had a history of being HIV positive, so the entire sample was retained for the secondary analysis.

The study population consisted of 376 (24.3%) 0-2-year-old and 1173 (75.7%) 3-5-year-old children. The total number of children with ECC was 66 (4.3%). Of these, eight (2.1%) 0–2 year-old children had ECC, and 58 (4.9%) 3-5-year-old children had ECC. The prevalence of ECC in the 3-5-year-old children was significantly higher than in the 0-2-year-old children ($\chi^2$ = 5.54; p = 0.019).

Of the 1549 children, 459 (29.6%) were wasted, 393 (25.4%) were stunted, 359 (23.2%) were underweight, and 142 (9.2%) were overweight. Of the 376 children who were 0–2 years old, 94 (25.0%) were wasted, 94 (25.0%) were stunted, 58 (14.4%) were underweight, and 66 (17.6%) were overweight. Of the 1173 children who were 3–5 years old, 405 (34.5%) were wasted, 299

(25.5%) were stunted, 301 (25.7%) were underweight, and 76 (6.5%) were overweight. There was a significant difference between the nutritional status of 0-2-year-old and 3-5-year-old children: the 3-5-year-old children were more likely to be underweight (PR = 1.66, p<0.001) and less likely to be overweight (PR = 0.36, p<0.001).

Of the 1549 children, 57 (3.7%) had enamel hypoplasia, 25 (1.6%) had hypomineralized second primary molar, ten (0.6%) had fluorosis, and 12 (0.8%) had amelogenesis imperfecta. Of the 376 children who were 0–2 years old, ten (2.7%) had enamel hypoplasia, none had hypomineralized second primary molar, two (0.5%) had fluorosis, and two (0.5%) had amelogenesis imperfecta. Of the 1173 children who were 3–5 years old, 47 (4.0%) had enamel hypoplasia, 25 (2.1%) had hypomineralized second primary molar, eight (0.7%) had fluorosis, and ten (0.9%) had amelogenesis imperfecta.

## ECC and malnutrition

Tables 1–3 highlight the nutritional status indicators that were associated with ECC in 0-2-year-old children, 3-5-year-old children, and 0–5 year-old children. Model 3 in each of the tables (fully adjusted) showed that underweight, stunting, and wasting/overweight were not significant risk indicators of ECC.

## ECC and enamel defects

Table 1, Model 2 shows that 0-2-year-old children who were wasted (p = 0.046), had hypoplasia (p = 0.044), fluorosis (p<0.001) and amelogenesis imperfecta (p<0.001) were significantly more likely to have ECC than were children who did not have these conditions. In model 3, which was adjusted for ECC risk factors and socio-economic status, only children with amelogenesis imperfecta (APR: 53.36; 95% CI: 11.63–244.78; p<0.001) and fluorosis (p<0.001) were significantly more likely to have ECC than children who did not have these lesions. Poor oral hygiene was also significantly associated with ECC (P< 0.001).

Table 2, Model 2 shows that 3-5-year-old children who were underweight were significantly less likely to have ECC than children who were not underweight (p = 0.027). Also, children who had hypoplasia (p = 0.013), hypomineralized second primary molars (p = 0.036), fluorosis (p<0.001), and amelogenesis imperfecta (p<0.001) were significantly more likely to have ECC than children who did not have these lesions. When the model was fully adjusted for ECC risk factors and socio-economic status (Model 3), children with hypoplasia (APR: 3.84; 95% CI: 1.60–9.19; p = 0.003), amelogenesis imperfecta (p<0.001) and fluorosis (p<0.001) were significantly more likely to have ECC than children who did not have these lesions. Children with poor oral hygiene were also almost three times more likely to have ECC than children with good oral hygiene (APR: 2.75; 95% CI: 1.14–6.64; p = 0.025).

In Table 3, Model 2, where data of the two age groups were combined, children who were underweight were significantly less likely to have ECC than children who were not underweight (p = 0.048). Also, children who had hypoplasia (p = 0.004), hypomineralized second primary molars (p = 0.016), and fluorosis (p<0.001) were significantly more likely to have ECC than children who did not have these lesions. When the analysis was adjusted for ECC risk factors and socio-economic status in Model 3, only children with hypoplasia (APR: 4.15; 95% CI: 1.96–8.80; p<0.001) and fluorosis (p<0.001) were significantly more likely to have ECC than children who did not have these lesions.

## Malnutrition and enamel defects

Table 4 illustrates the association between malnutrition and enamel defects. Overweight was significantly associated with amelogenesis imperfecta in the three age groups (P< 0.001).

**Table 1. Poisson regression model for factors associated with early childhood caries in 0-2-year-old children (N = 376).**

| Variables | Model 1 APR (95% CI) | p-value | Model 2 APR (95% CI) | p-value | Model 3 APR (95% CI) | p-value |
|---|---|---|---|---|---|---|
| **Underwight** | | | | | | |
| Not underweight | 1 | - | 1 | - | 1 | - |
| Underweight | 1.37(0.33–5.72) | 0.665 | 2.48(0.54–11.44) | 0.245 | 1.53(0.39–6.07) | 0.541 |
| **Stunted** | | | | | | |
| Not stunted | 1 | - | 1 | - | 1 | - |
| Stunted | 2.31(0.45–11.96) | 0.319 | 2.43(0.42–14.16) | 0.323 | 3.65(0.43–31.13) | 0.236 |
| **Wasted/Overweight** | | | | | | |
| Normal | 1 | - | 1 | - | 1 | - |
| Wasted | 5.77(0.91–36.46) | 0.062 | 6.34(1.04–38.68) | 0.046 | 5.57(0.70–44.41) | 0.105 |
| Overweight | 1.17(0.13–10.29) | 0.886 | 1.53(0.16–14.20) | 0.709 | 1.03(0.10–10.15) | 0.980 |
| **Enamel Defects** | | | | | | |
| No lesion | | | 1 | - | 1 | - |
| Enamel hypoplasia | | | 11.04(1.07–114.33) | 0.044 | 7.75(0.77–77.88) | 0.082 |
| Fluorosis | | | ** | <0.001 | ** | <0.001 |
| Amelogenesis imperfecta | | | 30.75(9.30–101.74) | <0.001 | 53.36(11.63–244.78) | <0.001 |
| **Oral Hygiene Status** | | | | | | |
| Good | | | | | 1 | - |
| Fair | | | | | 3.44(0.56–21.23) | 0.183 |
| Poor | | | | | ** | <0.001 |
| **Frequency of Daily Consumption of Sugar** | | | | | | |
| <3 times daily | | | | | 1 | - |
| ≥ 3 times daily | | | | | 1.07(0.26–4.45) | 0.924 |
| **Socio-Economic Status** | | | | | | |
| High | | | | | 1 | - |
| Middle | | | | | 1.28(0.29–5.63) | 0.747 |
| Low | | | | | 1.19(0.24–5.90) | 0.834 |
| **Constant** | 0.01(0.00–0.05) | <0.001 | 0.00 (0.00–0.03) | 0.000 | 0.00(0.00–0.01) | <0.001 |
| **Pseudo R²** | **0.09** | | **0.19** | | **0.23** | |

APR: adjusted prevalence ratio, CI: confidence interval.

**Extremely large or too small estimates due to condition prevalence

Model 1: malnutrition

Model 2: Model 1 + enamel defects

Model 3: Model 2 + oral hygiene, sugar consumption and socioeconomic status

Wasting and stunting in the 0-2-year-olds and combined age groups were significantly associated with fluorosis (P = 0.002 and P<0.001 respectively). Overweight was significantly associated with hypomineralized second molar in 3-5-year-old children and the combined age groups (P< 0.001respectively). Wasting was significantly associated with enamel hypoplasia in the 3-5-year-old children and the combined age groups (P = 0.047 and P = 0.037, respectively).

## Discussion

The study highlights three things. First, different types of malnutrition were associated with enamel defects, and enamel defects were associated with ECC. In the adjusted models simultaneously including all factors, enamel defects were the only factors significantly associated with ECC, and none of the various forms of malnutrition was a significant risk indicator for ECC. Second, the types of malnutrition and enamel defects associated with ECC differed for 0-

**Table 2. Poisson regression model for factors associated with early childhood caries in 3–5-year-old children (N = 1,173).**

| Variables | Model 1 APR (95% CI) | p-value | Model 2 APR (95% CI) | p-value | Model 3 APR (95% CI) | p-value |
|---|---|---|---|---|---|---|
| **Underweight** | | | | | | |
| Not underweight | 1 | - | 1 | - | 1 | - |
| Underweight | 0.37(0.15–0.95) | 0.038 | 0.34(0.13–0.89) | 0.027 | 0.38(0.14–1.05) | 0.062 |
| **Stunted** | | | | | | |
| Not stunted | 1 | - | 1 | - | 1 | - |
| Stunted | 1.25(0.59–2.65) | 0.569 | 1.34(0.62–2.87) | 0.456 | 1.15(0.50–2.64) | 0.743 |
| **Wasted/Overweight** | | | | | | |
| Normal | 1 | - | 1 | - | 1 | - |
| Wasted | 1.10(0.61–2.00) | 0.751 | 1.27(0.69–2.33) | 0.435 | 1.35(0.72–2.51) | 0.345 |
| Overweight | 0.38(0.09–1.58) | 0.183 | 0.43(0.10–1.77) | 0.241 | 0.52(0.12–2.21) | 0.376 |
| **Enamel Defects** | | | | | | |
| No lesion | | | 1 | - | 1 | - |
| Enamel hypoplasia | | | 3.13(1.27–7.69) | 0.013 | 3.84(1.60–9.19) | 0.003 |
| Fluorosis | | | 3.13(1.08–9.09) | 0.036 | 2.20(0.72–6.69) | 0.164 |
| Amelogenesis imperfecta | | | ** | <0.001 | ** | <0.001 |
| **Oral Hygiene Status** | | | | | | |
| Good | | | | | 1 | - |
| Fair | | | | | 0.96(0.51–1.78) | 0.888 |
| Poor | | | | | 2.75(1.14–6.64) | 0.025 |
| **Frequency of Daily Consumption of Sugar** | | | | | | |
| <3 times daily | | | | | 1 | - |
| ≥ 3 times daily | | | | | 1.47(0.80–2.72) | 0.217 |
| **Socio-Economic Status** | | | | | | |
| High | | | | | 1 | - |
| Middle | | | | | 1.20(0.60–2.43) | 0.606 |
| Low | | | | | 1.73(0.85–3.53) | 0.129 |
| **Constant** | 0.06(0.04–0.08) | <0.001 | 0.05 (0.03–0.08) | <0.001 | 0.03(0.02–0.07) | <0.001 |
| **Pseudo $R^2$** | **0.02** | | **0.05** | | **0.07** | |

PR: adjusted prevalence ratio, CI: confidence interval.

**Extremely large or too small estimates due to condition prevalence

Model 1: malnutrition

Model 2: Model 1 + enamel defects

Model 3: Model 2 + oral hygiene, sugar consumption and socioeconomic status

2-year-olds and 3-5-year-olds; merging the two age groups masks these differences. Third, though oral hygiene status and frequency of consumption of refined carbohydrate in-between-meals were associated with ECC, the effect of these factors was ameliorated by malnutrition and enamel defects, with oral hygiene being a significant risk indicator for ECC in 0-2-year-olds and 3-5-year-olds.

These findings confirm the association between malnutrition and enamel defects, and the association between enamel defects and ECC. The study highlights these association in a populations where ECC prevalence is low, thus complementing previous studies conducted among populations with high prevalence of ECC. The findings also reinforce the need to describe the epidemiological profile of ECC of 0-2-year-olds separately from that of 3-5-year-olds, and highlights the need to construct study models simulating real life to capture the effect of interacting factors on the risk for ECC.

**Table 3. Poisson regression model for factors associated with early childhood caries in 0-5-year-old children (N = 1,549).**

| Variables | Model 1 APR (95% CI) | p-value | Model 2 APR (95% CI) | p-value | Model 3 APR (95% CI) | p-value |
|---|---|---|---|---|---|---|
| **Underweight** | | | | | | |
| Not underweight | 1 | - | 1 | - | 1 | - |
| Underweight | 0.49(0.23–1.05) | 0.066 | 0.46(0.21–0.99) | 0.048 | 0.44(0.19–1.02) | 0.056 |
| **Stunted** | | | | | | |
| Not stunted | 1 | - | 1 | - | 1 | - |
| Stunted | 1.37(0.69–2.71) | 0.369 | 1.45(0.72–2.91) | 0.296 | 1.38(0.67–2.88) | 0.384 |
| **Wasted/Overweight** | | | | | | |
| Normal | 1 | - | 1 | - | 1 | - |
| Wasted | 1.36(0.78–2.36) | 0.282 | 1.59(0.91–2.80) | 0.105 | 1.66(0.92–2.99) | 0.094 |
| Overweight | 0.38(0.12–1.19) | 0.095 | 0.43(0.14–1.37) | 0.156 | 0.49(0.15–1.59) | 0.236 |
| **Enamel Defects** | | | | | | |
| No lesion | | | 1 | - | 1 | - |
| Enamel hypoplasia | | | 3.30(1.47–7.42) | 0.004 | 4.15(1.96–8.80) | <0.001 |
| Fluorosis | | | 3.46(1.26–9.49) | 0.016 | 2.38(0.84–6.75) | 0.103 |
| Amelogenesis imperfecta | | | ** | <0.001 | ** | <0.001 |
| **Oral Hygiene Status** | | | | | | |
| Good | | | | | 1 | - |
| Fair | | | | | 1.10(0.63–1.91) | 0.744 |
| Poor | | | | | 2.16(0.84–5.56) | 0.110 |
| **Frequency of Daily Consumption of Sugar** | | | | | | |
| <3 times daily | | | | | 1 | - |
| ≥ 3 times daily | | | | | 1.51(0.81–2.81) | 0.200 |
| **Socio-Economic Status** | | | | | | |
| High | | | | | 1 | - |
| Middle | | | | | 1.12(0.58–2.88) | 0.736 |
| Low | | | | | 1.57(0.80–3.11) | 0.191 |
| **Constant** | 0.04(0.03–0.06) | <0.001 | 0.04 (0.03–0.06) | 0.000 | 0.03(0.01–0.05) | <0.001 |
| **Pseudo R$^2$** | **0.01** | | **0.05** | | **0.07** | |

APR: adjusted prevalence ratio, CI: confidence interval.

**Extremely large or too small estimates due to condition prevalence

Model 1: malnutrition

Model 2: Model 1 + enamel defects

Model 3: Model 2 + oral hygiene, sugar consumption and socioeconomic status

The study highlights that ECC is a multifactorial disease, resulting from complex interactions between several factors that differ according to age. Nigeria has a low prevalence of ECC [50], a high prevalence of malnutrition [4], and a high prevalence of hypoplasia and hypomineralized primary second molars [19–21]. The study suggests that with poor oral hygiene practices, the rough pitted surfaces of defective enamel in amelogenesis imperfecta and fluorosis have higher risk of plaque retention and in-turn, higher risk for ECC. Though malnutrition does not have a direct association with ECC, its association with enamel defects makes it a distal risk factor for ECC. Past studies have indicated that malnutrition significantly increases the risk of hypoplasia [12, 48] and may increase the risk of hypomineralized primary second molars [51].

There are no prior studies on the association between malnutrition, amelogenesis imperfecta and fluorosis. The plausibility of such an association seems remote since amelogenesis

**Table 4. Poisson regression model for association between enamel defects and malnutrition (N = 1420).**

| Variables | Enamel hypoplasia APR (95% CI) | p-value | Hypomineralized primary second molar APR (95% CI) | p-value | Fluorosis APR (95% CI) | p-value | Amelogenesis imperfecta APR (95% CI) | p-value |
|---|---|---|---|---|---|---|---|---|
| **0-2-year-olds** | | | | | | | | |
| Normal | 1.00 | - | - | - | 1.00 | - | 1.00 | - |
| Underweight | ** | <000.1 | - | - | 30.38(0.68—**) | 0.078 | ** | <0.001 |
| Stunted | 1.29(0.23–7.15) | 0.771 | - | - | 0.33(0.17–0.66) | 0.002 | ** | <0.001 |
| Wasted | 0.51(0.06–4.18) | 0.531 | - | - | ** | <0.001 | 2.41(0.15–38.07) | 0.531 |
| Overweight | 1.20(0.22–6.66) | 0.834 | - | - | 7.32(0.15–353.04) | 0.314 | ** | <0.001 |
| **3-5-year-olds** | | | | | | | | |
| Normal | 1.00 | - | 1.00 | - | 1.00 | - | 1.00 | - |
| Underweight | 1.87(0.41–1.86) | 0.721 | 0.56(0.13–2.48) | 0.445 | ** | 0.995 | 2.06(0.53–7.95) | 0.294 |
| Stunted | 1.11(0.57–2.17) | 0.754 | 0.74(0.24–2.31) | 0.608 | ** | 0.996 | 0.38(0.10–1.45) | 0.156 |
| Wasted | 0.41(0.17–0.99) | 0.047 | 0.27(0.05–1.38) | 0.117 | ** | 0.997 | ** | <0.001 |
| Overweight | 0.23(0.03–1.60) | 0.136 | ** | <0.001 | ** | 0.999 | ** | <0.001 |
| **0-5-year-olds** | | | | | | | | |
| Normal | 1.00 | - | 1.00 | - | 1.00 | - | 1.00 | - |
| Underweight | 0.76(0.36–1.64) | 0.490 | 1.37(0.40–4.72) | 0.618 | 3.81(0.42–34.34) | 0.331 | 0.96(0.14–6.38) | 0.965 |
| Stunted | 1.12(0.58–2.17) | 0.745 | 0.44(0.11–1.69) | 0.231 | 0.22(0.11–0.41) | <0.001 | 0.47(0.11–1.97) | 0.303 |
| Wasted | 0.42(0.19–0.95) | 0.037 | 0.41(0.14–1.21) | 0.107 | ** | <0.001 | 0.30(0.03–3.50) | 0.338 |
| Overweight | 0.55(0.18–1.62) | 0.277 | ** | <0.001 | 2.61(0.24–28.41) | 0.431 | ** | <0.001 |

APR: adjusted prevalence ratio, CI: confidence interval.

**Extremely large or too small estimates due to condition prevalence

imperfecta is a genetic disorder, whereas fluorosis results from fluoride overdose [52]. There is, however, evidence that malnutrition in the first five years of life may be a risk factor for genetic expression of diseases [53]: nutrients can directly inhibit epigenetic enzymes or alter the availability of substrate needed for enzymatic reactions [54, 55]. Nutrition may also increase the susceptibility of developing teeth to toxins, as indicated by studies showing that fluorosis in the primary dentition is not exclusively dependent on water fluoride concentrations [56].

Some strategies that can reduce the risk of ECC in the study population include improving the quality and frequency of tooth brushing [57] along with the use of fluoridated toothpaste [58] to reduce plaque retention and increase the exposure of oral biofilm to fluoride ions. Another strategy may be through reducing the risk of malnutrition although we cannot provide conclusive evidence. Although we found significant associations between the various forms of malnutrition and enamel defects, the low prevalence of some forms of enamel defects in the study population made it difficult to determine the direction of associations and we are, therefore, unable to show if reducing the risk of malnutrition may reduce the risk of enamel defects.

This study highlighted unusual findings that raise questions: Unlike past studies that found that hypomineralized second primary molar [20, 21] and frequency of sugar consumption [27] were risk indicators for ECC, we found no such association when nutritional status was controlled for. This finding is inconsistent with the assumption that ECC is an exclusive sugar-related disease [41]. A possible explanation for our finding may be that poor oral hygiene, malnutrition and high consumption of free sugar share a common risk indicator (low socio-economic status), which may have offset the association between sugar consumption and ECC.

Further studies are needed to elucidate the complex relationships and interactions among risk factors for ECC.

This study also highlights the need to disaggregate ECC data by age: The epidemiological profile of ECC for 0-2-year-old children differed from that of 3-5-year-old children. We also showed that not splitting the age groups could mask information that is important for understanding the epidemiology of ECC. For this study, the risk indicators for ECC in 0-5-year-old children mirrored more the profile of ECC in 3-5-year-old children than of ECC in 0-2-year-old children. This finding reinforces the need to conduct epidemiological studies on ECC for the two age groups– 0-2-year-olds and 3-5-year-olds–separately, a view shared with El Tantawi et al [49].

The study had limitations. First, it was cross-sectional so causal inferences cannot be deduced from the findings. However, we controlled for several variables that are known to be associated with ECC, thereby providing evidence that can be explored in further research. The findings also highlight the need for studies that show how risk factors interact to predispose individuals to ECC. Second, the original sample was planned to assess the relationship between ECC and maternal psychosocial factors and might have lacked adequate power to support analyses for the association between ECC and enamel defects; the wide confidence intervals imply that some of the findings should be taken with caution. Third, the study was conducted in one of the 774 local government areas in the country, so the findings cannot be generalized to all of Nigeria.

## Conclusions

This study was conducted in a sub-urban Nigerian population with high prevalence of malnutrition and enamel defects but with low prevalence of ECC. It revealed that various types of malnutrition were associated with various forms of enamel defects. Though enamel defects were a risk indicator for ECC, malnutrition was not significantly associated with ECC in fully adjusted models. Only oral hygiene was a risk indicator for ECC after nutritional status and enamel defects were considered. The findings from this study should be explored further.

## Acknowledgments

We acknowledge and thank the study participants for the contributions they made to generating new knowledge. We also thank Prof. Carlos A. Feldens who shared insights with us on the study findings.

## Author Contributions

**Conceptualization:** Morenike Oluwatoyin Folayan, Abiola Adeniyi, Tracy L. Finlayson.

**Data curation:** Morenike Oluwatoyin Folayan, Ayodeji Babatunde Oginni, Michael Alade.

**Formal analysis:** Morenike Oluwatoyin Folayan, Maha El Tantawi, Ayodeji Babatunde Oginni.

**Investigation:** Morenike Oluwatoyin Folayan, Michael Alade.

**Methodology:** Morenike Oluwatoyin Folayan, Maha El Tantawi, Ayodeji Babatunde Oginni, Michael Alade, Abiola Adeniyi.

**Project administration:** Morenike Oluwatoyin Folayan.

**Resources:** Morenike Oluwatoyin Folayan, Michael Alade.

**Supervision:** Morenike Oluwatoyin Folayan.

**Writing – original draft:** Morenike Oluwatoyin Folayan.

**Writing – review & editing:** Morenike Oluwatoyin Folayan, Maha El Tantawi, Ayodeji Babatunde Oginni, Michael Alade, Abiola Adeniyi, Tracy L. Finlayson.

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
