## [Decision Letter · Decision Letter 0]

14 Apr 2020

PONE-D-20-05626

Malnutrition, Enamel Defects, and Early Childhood Caries in Preschool Children in a Sub-Urban Nigeria Population

PLOS ONE

Dear Dr Folayan,

Thank you for submitting your manuscript to PLOS ONE. After careful consideration, we feel that it has merit but does not fully meet PLOS ONE’s publication criteria as it currently stands. Therefore, we invite you to submit a revised version of the manuscript that addresses the points raised during the review process.

We would appreciate receiving your revised manuscript by May 29 2020 11:59PM. To enhance the reproducibility of your results, we recommend that if applicable you deposit your laboratory protocols in protocols.io, where a protocol can be assigned its own identifier (DOI) such that it can be cited independently in the future. For instructions see: http://journals.plos.org/plosone/s/submission-guidelines#loc-laboratory-protocols

We look forward to receiving your revised manuscript.

Kind regards,

Denis Bourgeois

Academic Editor

PLOS ONE

Journal Requirements:

Additional Editor Comments (if provided):

It is necessary to have precise information on the scientific added value and the real contribution made by the submission.

Reviewers' comments:

Reviewer's Responses to Questions

**Comments to the Author**

1. Is the manuscript technically sound, and do the data support the conclusions?

Reviewer #1: Yes

2. Has the statistical analysis been performed appropriately and rigorously? 

Reviewer #1: Yes

3. Have the authors made all data underlying the findings in their manuscript fully available?

Reviewer #1: Yes

4. Is the manuscript presented in an intelligible fashion and written in standard English?

Reviewer #1: Yes

5. Review Comments to the Author

Reviewer #1: Minor essential revision:

The paper is not an original article due to the fact that it is a secondary analysis of primary data. This should be clearly stated.

Please add the original paper in the reference list.

The reference of the questionnaire is lacking. Please add in the reference list.

No information about calibration of examiners is provided. This should be added.

Why was dmft for the analysis used and no acutal disease score (d)?

The definition of ECC that is used in the paper is quite outdated, what is the decision based on that you used it?

6. PLOS authors have the option to publish the peer review history of their article (what does this mean?). If published, this will include your full peer review and any attached files.

Reviewer #1: No

---

## [Author Response · Author response to Decision Letter 0]

16 Apr 2020

http://www.journals.plos.org/plosone/s/file?id=wjVg/PLOSOne_formatting_sample_main_body.pdf

• RESPONSE: The manuscript has been revised in line with the guidelines. The headings have been revised and the tables redrawn.

http://www.journals.plos.org/plosone/s/file?id=ba62/PLOSOne_formatting_sample_title_authors_affiliations.pdf

• RESPONSE: The title page has been edited in line with the guidelines. 

It is necessary to have precise information on the scientific added value and the real contribution made by the submission.

• RESPONSE: Thanks for highlighting this. We have included statements to describe what we consider to be the scientific added value and the real contribution made by the submission. Please see lines 329-335.

The paper is not an original article due to the fact that it is a secondary analysis of primary data. This should be clearly stated.

• RESPONSE: Thanks for raising this. This is stated in the abstract. It is also stated in the line 101 of the methods section. We have reiterated this information throughout the methods section; please see lines 109, 117, 122, 140. Lines 385-388 in the section on study limitation also reiterates this information. We also checked the entire file using Ctrl+F to make sure the word “original” was not mentioned anywhere so that readers are not misled. It may be listed under the journal sections/ type of papers or studies as original study as opposed to a review/ systematic review, which in this context would be true. 

Please add the original paper in the reference list.

• RESPONSE: The original publication has been included as reference 30.

The reference of the questionnaire is lacking. Please add in the reference list.

• RESPONSE: The reference to the questionnaire used is reference 27: Khami MR, Virtanen JI, Jafarian M, Murtomaa H: Oral health behaviour and its determinants amongst Iranian dental students. Eur J Dent Educ. 2007, 11: 42-47.

No information about calibration of examiners is provided. This should be added.

• RESPONSE: This information was provided in the methods section of the manuscript – lines 140-146.

Why was dmft for the analysis used and no actual disease score (d)?

RESPONSE: The formal definition of Early Childhood Caries includes decayed, filled or missing surfaces in any tooth in children younger than 6 years (http://abodontopediatria.org.br/site/wp-content/uploads/2019/05/ipd.12490.pdf) so we used the dmft and not just the (d). 

The definition of ECC that is used in the paper is quite outdated, what is the decision based on that you used it?

• RESPONSE: Thanks for raising this point. We have revised using the current version of the definition as reported in the IAPD Bangkok declaration (http://abodontopediatria.org.br/site/wp-content/uploads/2019/05/ipd.12490.pdf) and Tinanoff N, Baez RJ, Guillory CD, Donly KJ, Feldens CA, McGrath C, et al. Early childhood caries epidemiology, aetiology, risk assessment, societal burden, management, education, and policy: Global perspective. Int J Paediatr Dent. 2019; 29:238–248.

---

## [Decision Letter · Decision Letter 1]

28 Apr 2020

Malnutrition, enamel defects, and early childhood caries in preschool children in a sub-urban Nigeria population

PONE-D-20-05626R1

Dear Dr. Folayan,

We are pleased to inform you that your manuscript has been judged scientifically suitable for publication and will be formally accepted for publication once it complies with all outstanding technical requirements.

With kind regards,

Denis Bourgeois

Academic Editor

PLOS ONE

Additional Editor Comments (optional):

Reviewers' comments:

Reviewer's Responses to Questions

**Comments to the Author**

1. If the authors have adequately addressed your comments raised in a previous round of review and you feel that this manuscript is now acceptable for publication, you may indicate that here to bypass the “Comments to the Author” section, enter your conflict of interest statement in the “Confidential to Editor” section, and submit your "Accept" recommendation.

Reviewer #1: All comments have been addressed

2. Is the manuscript technically sound, and do the data support the conclusions?

Reviewer #1: Yes

3. Has the statistical analysis been performed appropriately and rigorously? 

Reviewer #1: Yes

4. Have the authors made all data underlying the findings in their manuscript fully available?

Reviewer #1: Yes

5. Is the manuscript presented in an intelligible fashion and written in standard English?

Reviewer #1: Yes

6. Review Comments to the Author

Reviewer #1: The authors have adequately addressed the comments raised in a previous round of review and i feel that this manuscript is now acceptable for publication.

7. PLOS authors have the option to publish the peer review history of their article (what does this mean?). If published, this will include your full peer review and any attached files.

Reviewer #1: No

---

## [Editor Report · Acceptance letter]

12 Jun 2020

PONE-D-20-05626R1 

Malnutrition, enamel defects, and early childhood caries in preschool children in a sub-urban Nigeria population 

Dear Dr. Folayan:

I'm pleased to inform you that your manuscript has been deemed suitable for publication in PLOS ONE. Congratulations! Your manuscript is now with our production department. 

Kind regards, 

on behalf of

Professor Denis Bourgeois 

Academic Editor

PLOS ONE